# Long-Time Coherent Integration Method for Passive Bistatic Radar Using Frequency Hopping Signals

**DOI:** 10.3390/s24196236

**Published:** 2024-09-26

**Authors:** Gang Chen, Xiaowei Biao, Yi Jin, Changzhi Xu, Yifan Ping, Sujun Wang

**Affiliations:** China Academy of Space Technology, Xi’an 710100, China; biaoxw123@163.com (X.B.); john.0216@163.com (Y.J.); sandy_xu@126.com (C.X.); pingyf1982@126.com (Y.P.); wangsujun504@163.com (S.W.)

**Keywords:** coherent integration, frequency hopping, range walk, range curve, Doppler frequency migration, passive bistatic radar

## Abstract

Long-time coherent integration using frequency hopping signals is a challenging problem for passive bistatic radar due to its frequency hopping characteristics. Apart from range walk, range curve, and Doppler frequency migration, Doppler diffusion caused by frequency hopping characteristics occurs within the observation time, which also lowers the detection performance. To deal with this problem, a novel coherent integration method for frequency hopping signals based on passive bistatic radar is proposed in this paper. In this novel method, range curve and range walk are eliminated by applying generalized Keystone transform. Then, Doppler frequency migration caused by the target’s acceleration is compensated for by a parameter search with a designed search scope. Finally, Doppler frequency migration caused by frequency hopping characteristics is compensated for by designing a new acceleration compensation function and a revised rotation factor for Fourier transform. Since migration effects caused by frequency hopping characteristics are considered and compensated for when using frequency hopping signals, the weak target echo can be better integrated in the observation time compared to when using the existing methods. The simulation results and performance analysis illustrate the effectiveness of the proposed method.

## 1. Introduction

As a special case of bistatic radar, passive bistatic radar (PBR) contains no co-located transmitter but utilizes electromagnetic signals reflected by the target from the cooperative/non-cooperative illuminator of opportunity for target location and tracking [1,2,3]. Due to the silent working pattern, PBR has advantages in countering electronic interference and anti-radiation missiles. Additionally, the illuminator of opportunity that PBR uses usually occupies the low-frequency band and mainly covers low-altitude areas, which is beneficial for stealth and low-altitude target detection. These cooperative/non-cooperative illuminators include frequency modulation (FM) [4,5,6], digital video broadcasting–terrestrial/satellite (DVB-T/S) [7,8,9,10], digital audio broadcasting (DAB) [11,12], long-term evolution (LTE) [13,14], global navigation satellite systems (GNSS) [15,16], Inmarsat [17], Starlink [18], and so on.

These illuminators of opportunity usually transmit continuous wave signals with a low power level, and signal returns from the moving target are very weak. Additionally, targets with a small radar cross-section (RCS) also bring challenges to radars. Increasing the observation time can effectively raise the energy level of the weak target echo [19,20]. However, the energy level of the weak target echo is not always increased by using a long-time integration operation. The complex motion of the target lowers the integration performance. According to the motion of the moving target, three issues should be noted: (1) range walk (RW), which is caused by the velocity of the moving target; (2) range curve (RC) and Doppler frequency migration (DFM), which are caused by the acceleration of the moving target; and (3) irregular motion, which is caused by the high-order maneuverability of the moving target.

To solve the long-time integration problems mentioned above, many algorithms have been proposed. According to the target motion model, these algorithms can be divided into three categories: (1) Constant velocity model: The velocity of the target stays constant for the whole observation time. Related algorithms include Keystone transform (KT) [21,22], Radon Fourier transform (RFT) [23,24], and so on [25,26]. (2) Constant acceleration model: The acceleration of the target stays constant for the whole observation time. The main algorithms include generalized Keystone transform (GKT) [27,28], Radon fraction Fourier transform (RFRFT) [29,30], and so on [31,32,33]. (3) Jerk model: Both the velocity and the acceleration of the target vary with time. Research points focus on the Keystone transform and generalized de-chirp process (KTGDP) [34], generalized Radon Fourier Transform (GRFT) [35], and so on [36,37].

In the algorithms mentioned above, the carrier frequency of the transmitting signal is considered to stay constant for the observation time. However, for frequency hopping (FH) signals [38], the carrier frequency changes with time, which results in the Doppler frequency sensitivity of the transmitted signal. Thus, apart from RC, RW, and DFM, another DFM caused by frequency hopping characteristics will also lower the detection performance of the radar system [39]. Additionally, hopping frequency coupled with RC, RW, and DFM brings more serious cross-resolution cell accumulation problems when using the conventional method. To deal with the long-time integration problem for FH signals, segmented incoherent processing is usually applied. In reality, the integration gain in incoherent processing is lower than that in coherent processing. Thus, incoherent processing is not the optimal choice. To deal with this problem, a long-time coherent integration method for FH signals is proposed in this paper. In this novel method, RC and DFM are eliminated in the first step. RW is eliminated in the second step. DFM caused by the FH characteristic is compensated for in the last step. Since RC, RW, and two kinds of DFM are eliminated and compensated for, the weak target echo can be integrated coherently.

The rest of this paper is organized as follows. In Section 2, a signal model is established. The proposed method is described in detail in Section 3. The simulation results and performance analysis are presented in Section 4. Conclusions are presented in Section 5.

## 2. Signal Model

A typical PBR system geometry is shown in Figure 1.

From Figure 1, it can be seen that two sets of antenna are located near the terminal processor. One is the reference antenna which is used to receive the direct signal from the illuminator of opportunity for clutter cancellation and range compression. The other is the surveillance antenna which is applied to collect target echo from the surveillance area, but it is usually contaminated by the direct signal and scattered signal from the ground objects.

Supposing that the FH pulse signals are transmitted from the illuminator of opportunity, which can be represented as
(1)$$s_{\mathrm{ref}}\left(t\right)=d\left(t\right)\exp\left(j2\pi \left(f_{c}+\Delta f_{m}\right)t\right)$$
where *d*(*t*) is the complex envelope of the transmitting signal, *f*_c_ is the carrier frequency of the first pulse signal, and Δ*f_m_* represents the frequency difference between the *m*-th and the first pulse echo.

The baseband signals received by the surveillance channel after clutter cancellation [40] can be represented as
(2)$$s_{\mathrm{sur}}\left(t,t_{m}\right)=\sigma d\left(t-\frac{R\left(t_{m}\right)}{c}\right)\exp\left(-j2\pi \frac{R\left(t_{m}\right)}{c}\left(f_{c}+\Delta f_{m}\right)\right)$$
where σ denotes the target reflectivity. The target reflectivity is usually considered to be unchanged during the integration time, σ is a constant value. *c* is the velocity of light, *t* and *t_m_* denote the fast time and the slow time, respectively, and *R*(*t_m_*) represents the instantaneous range between the radar and the target, which is represented as
(3)$$R\left(t_{m}\right)=R_{0}+vt_{m}+\frac{1}{2}at_{m}^{2}$$
where *R*_0_, *v*, and *a* represent the target’s initial range, velocity, and acceleration, respectively.

When applying Fourier transform (FT) to Equation (2) with respect to fast time *t*, the surveillance signal in the range–frequency domain can be written as
(4)$$S_{\mathrm{sur}}\left(f,t_{m}\right)=\sigma D\left(f\right)\exp\left(-j2\pi \frac{R\left(t_{m}\right)}{c}\left(f+f_{c}+\Delta f_{m}\right)\right)$$
where *D*(*f*) represents the FT form of *d*(*t*).

Then, when applying range compression in the fast time frequency domain, Equation (4) can be represented as
(5)$$S\left(f,t_{m}\right)=\sigma {\left|D\left(f\right)\right|}^{2}\exp\left(-j2\pi \frac{R\left(t_{m}\right)}{c}\left(f+f_{c}+\Delta f_{m}\right)\right)$$

Substituting Equation (3) into Equation (5), we obtain
(6)$$\begin{array}{cc} S\left(f,t_{m}\right) & =\sigma_{s}{\left|D\left(f\right)\right|}^{2}\exp\left(-j2\pi \frac{R_{0}}{c}\left(f_{c}+\Delta f_{m}\right)\right)\exp\left(-j2\pi \frac{R_{0}}{c}f\right)\exp\left(-j2\pi \frac{vt_{m}}{c}\left(f_{c}+f\right)\right)\\ & \cdot \exp\left(-j2\pi \frac{at_{m}^{2}}{2c}\left(f_{c}+f\right)\right)\exp\left(-j2\pi \frac{vt_{m}}{c}\Delta f_{m}\right)\exp\left(-j2\pi \frac{at_{m}^{2}}{2c}\Delta f_{m}\right) \end{array}$$

From the third and the fourth phase items in Equation (6), it can be seen that fast time frequency *f* is coupled with slow time *t_m_*, which results in RW and RC. From the fourth phase item, it is also known that DFM occurs due to the second-order phase item caused by the target’s acceleration. The fifth and sixth phase items show the first-order and second-order DFMs caused by the FH characteristics.

To sum up, RC, RW, and two kinds of DFM will lower the detection performance of the radar system. To solve this problem, a novel long-time coherent integration method for FH signals is proposed and discussed in the next section.

## 3. Proposed Method

In this section, the proposed method is introduced, and its flowchart is shown in Figure 2.

In Figure 2, GKT is applied to eliminate RW and RC. DFM caused by the target’s acceleration is compensated for by a parameter search with the designed search scope, whereas DFM caused by the FH characteristics is compensated for by designing a new acceleration compensation function and a revised rotation factor for FT. For convenience, DFM caused by the target’s acceleration is called DFM-I, whereas DFM caused by the FH characteristics is called DFM-II in the following discussion.

### 3.1. Range Curve Correction

GKT was applied to remove RC caused by the target’s constant acceleration, which performs scaling *t_m_* = [*f*_c_/(*f* + *f*_c_)]^1/2^*τ_m_* in the *t*_m_-*f* domain. Substituting the scaling factor into Equation (6), we obtain
(7)$$\begin{array}{ll} S\left(f,\tau_{m}\right) & =\sigma_{s}{\left|D\left(f\right)\right|}^{2}\exp\left(-j2\pi \frac{R_{0}}{c}f\right)\exp\left(-j2\pi \frac{v}{c}\tau_{m}\sqrt{f_{c}\left(f_{c}+f\right)}\right)\exp\left(-j2\pi \frac{a}{2c}\tau_{m}^{2}f_{c}\right)\\ & \cdot \exp\left(-j2\pi \frac{v}{c}\tau_{m}\sqrt{\frac{f_{c}}{f_{c}+f}}\Delta f_{m}\right)\exp\left(-j2\pi \frac{a}{2c}\tau_{m}^{2}\frac{f_{c}}{f_{c}+f}\Delta f_{m}\right) \end{array}$$
where σ_s_ denotes a constant value, which is written as
(8)$$\sigma_{s}=\sigma \exp\left(-j2\pi \frac{R_{0}}{c}\left(f_{c}+\Delta f_{m}\right)\right)$$

In Equation (7), it can be seen that the second-order coupling term has been removed, whereas the first-order coupling still exists.

### 3.2. Doppler Frequency Migration-I Correction

After RC correction, the third phase item in Equation (7), which is induced by the target’s acceleration, needs to be estimated and compensated for before RW correction. The estimated value *a*_0_ is solved by the cost function below.
(9)$$a_{0}=\arg\max\left|S\left(f,\tau_{m}\right)\cdot H\left(\tau_{m}\right)\right|$$
where *H*(*τ_m_*) is the acceleration correction factor, which is written as
(10)$$H\left(\tau_{m}\right)=\exp\left(j2\pi \frac{a_{0}}{2c}\tau_{m}^{2}f_{c}\right)$$

Compensating for Equation (7) with Equation (10), we obtain
(11)$$\begin{array}{ll} S_{a}\left(f,\tau_{m}\right) & =\sigma_{s}{\left|D\left(f\right)\right|}^{2}\exp\left(-j2\pi \frac{R_{0}}{c}f\right)\exp\left(-j2\pi \frac{v}{c}\tau_{m}\sqrt{f_{c}\left(f_{c}+f\right)}\right)\exp\left(-j2\pi \frac{\left(a-a_{0}\right)}{2c}\tau_{m}^{2}f_{c}\right)\\ & \cdot \exp\left(-j2\pi \frac{v}{c}\tau_{m}\sqrt{\frac{f_{c}}{f_{c}+f}}\Delta f_{m}\right)\exp\left(-j2\pi \frac{a}{2c}\tau_{m}^{2}\frac{f_{c}}{f_{c}+f}\Delta f_{m}\right) \end{array}$$

When *a* = *a*_0_, *S_a_*(*f*, *τ_m_*) reaches its maximum value. Equation (11) can be further represented as
(12)$$\begin{array}{ll} S_{a}\left(f,\tau_{m}\right) & =\sigma_{s}{\left|D\left(f\right)\right|}^{2}\exp\left(-j2\pi \frac{R_{0}}{c}f\right)\exp\left(-j2\pi \frac{v}{c}\tau_{m}\sqrt{f_{c}\left(f_{c}+f\right)}\right)\exp\left(-j2\pi \frac{\left(a-a_{0}\right)}{2c}\tau_{m}^{2}f_{c}\right)\\ & \cdot \exp\left(-j2\pi \frac{v}{c}\tau_{m}\sqrt{\frac{f_{c}}{f_{c}+f}}\Delta f_{m}\right)\exp\left(-j2\pi \frac{a}{2c}\tau_{m}^{2}\frac{f_{c}}{f_{c}+f}\Delta f_{m}\right) \end{array}$$

In Equation (12), it is observed that the DFM effect caused by the target’s acceleration is compensated for. Then, the RW effect needs to be corrected and is discussed in the next subsection.

### 3.3. Range Walk Correction

Although RC and DFM have been corrected and compensated for, the target’s energy is still distributed into several range cells due to the RW effect. Thus, RW correction processing is required and introduced in this subsection.

GKT is applied to remove RW caused by the target’s constant velocity, which performs scaling *p_m_* = [*f*_c_/(*f* + *f*_c_)]^1/2^*τ_m_* in the *τ_m_*-*f* domain. Substituting the scaling factor into Equation (12), we obtain
(13)$$\begin{array}{ll} S_{a}\left(f,p_{m}\right) & =\sigma_{s}{\left|D\left(f\right)\right|}^{2}\exp\left(-j2\pi \frac{R_{0}}{c}f\right)\exp\left(-j2\pi \frac{v}{c}p_{m}f_{c}\right)\\ & \cdot \exp\left(-j2\pi \frac{v}{c}p_{m}\frac{f_{c}}{f_{c}+f}\Delta f_{m}\right)\exp\left(-j2\pi \frac{a}{2c}p_{m}^{2}{\left(\frac{f_{c}}{f_{c}+f}\right)}^{2}\Delta f_{m}\right) \end{array}$$

By converting Equation (13) to the *p_m_*-*t* domain form, it is represented as
(14)$$\begin{array}{ll} s_{a}\left(t,p_{m}\right) & =\sigma_{s}\sin c\left[\left(t-\frac{R_{0}}{c}\right)\right]\exp\left(-j2\pi \frac{v}{c}p_{m}f_{c}\right)\exp\left(-j2\pi \frac{v}{c}p_{m}\frac{f_{c}}{f_{c}+f}\Delta f_{m}\right)\\ & \cdot \exp\left(-j2\pi \frac{a}{2c}p_{m}^{2}{\left(\frac{f_{c}}{f_{c}+f}\right)}^{2}\Delta f_{m}\right) \end{array}$$

In Equation (14), it is observed that the envelope shift has been corrected. Then, when applying FT to transform Equation (14) into the slow time frequency domain, it is written as
(15)$$\begin{array}{ll} s_{a}\left(t,p_{m}\right) & =\int_{p_{m}}\sigma_{s}\sin c\left[\left(t-\frac{R_{0}}{c}\right)\right]\exp\left(-j2\pi \frac{v}{c}p_{m}f_{c}\right)\exp\left(-j2\pi \frac{v}{c}p_{m}\frac{f_{c}}{f_{c}+f}\Delta f_{m}\right)\\ & \cdot \exp\left(-j2\pi \frac{a}{2c}p_{m}^{2}{\left(\frac{f_{c}}{f_{c}+f}\right)}^{2}\Delta f_{m}\right)\exp\left(-j2\pi fp_{m}\right)dp_{m} \end{array}$$

It is known that when the phase item equals zeros, Equation (15) reaches its maximum value. It can be solved as
(16)$$v_{0}=-\frac{c}{f_{c}}f\frac{f_{c}+f}{f_{c}+f+\Delta f_{m}}-\frac{a}{2}p_{m}\frac{f_{c}}{f_{c}+f}\cdot \frac{\Delta f_{m}}{f_{c}+f+\Delta f_{m}}$$

In Equation (16), it can be seen that the estimated target velocity *v*_0_ is relevant to the hopping frequency Δ*f*_m_, the target’s acceleration *a*, and the slow time *p_m_*, which is not a constant value. The target’s energy diffuses into serval range and Doppler cells. Thus, DFM caused by the FH characteristics requires compensation.

### 3.4. Doppler Frequency Migration-II Correction

In this subsection, DFM caused by the FH characteristics is discussed and analyzed. In Equation (16), it can be seen that the velocity *v* and acceleration *a*, coupled with hopping frequency, results in DFM-II. This is a two-dimensional parameter estimation problem. Searching for two parameters simultaneously consumes plenty of computation resources. Thus, in this paper, DFM-II caused by the velocity is compensated for in each Doppler channel according to the relationship between the velocity and Doppler frequency, whereas the DFM-II caused by the acceleration is compensated for by searching for the acceleration *a*. In real-world applications, the research scope of acceleration can be set to a longer step size to meet more target acceleration due to the poor Doppler resolution.

As is well known, the Doppler frequency resolution ratio is inversely proportional to the total observation time *T*. Thus, the Doppler frequency in each Doppler channel can be represented as
(17)$$f_{k}=\frac{1}{T}\left(k-\frac{N}{2}\right)$$
where *k* is the index of the Doppler channel, and *N* is the total number of Doppler channels.

According to the relationship between the velocity and Doppler frequency, the velocity in each Doppler channel can be represented as
(18)$$v_{k}=\frac{f_{k}}{f_{c}}c=\frac{1}{T}\left(k-\frac{N}{2}\right)\frac{c}{f_{c}}$$

The compensation factor for DFM-II caused by the target’s velocity is written as
(19)$$H_{v}\left(p_{m},k\right)=\exp\left(j2\pi \frac{v_{k}}{c}p_{m}\frac{f_{c}}{f_{c}+f}\Delta f_{m}\right)$$

When applying the velocity compensation factor to the rotation factor of FT, it is written as
(20)$$\begin{array}{ll} H_{w}\left(p_{m},k\right) & =\exp\left(-j2\pi fp_{m}\right)\exp\left(j2\pi \frac{v_{k}}{c}p_{m}\frac{f_{c}}{f_{c}+f}\Delta f_{m}\right)\\ & =\exp\left(j2\pi p_{m}\left(\frac{v_{k}}{c}\Delta f_{m}\frac{f_{c}}{f_{c}+f}-f\right)\right) \end{array}$$

The compensation factor for DFM-II caused by the target’s acceleration is written as
(21)$$H_{a}\left(p_{m}\right)=\exp\left(j2\pi \frac{a_{0}}{2c}p_{m}{\left(\frac{f_{c}}{f_{c}+f}\right)}^{2}\Delta f_{m}\right)$$

When compensating for Equation (14) with Equation (21) and transforming it into the slow time frequency domain with a revised rotation factor in Equation (20), we obtain
(22)$$\begin{array}{ll} S_{a}\left(t,P_{m}\right) & =\arg\max\left|\int_{p_{m}}\left[\sigma_{s}\sin c\left[\left(t-\frac{R_{0}}{c}\right)\right]\exp\left(-j2\pi \frac{v}{c}p_{m}f_{c}\right)\exp\left(-j2\pi \frac{v}{c}p_{m}\frac{f_{c}}{f_{c}+f}\Delta f_{m}\right)\right.\right.\\ & \left.\left.\cdot \exp\left(-j2\pi \frac{a}{2c}p_{m}^{2}{\left(\frac{f_{c}}{f_{c}+f}\right)}^{2}\Delta f_{m}\right)\right]\cdot H_{a}\left(p_{m}\right)\cdot H_{w}\left(p_{m},k\right)dp_{m}\right| \end{array}$$

Furthermore, Equation (22) can be written as
(23)$$\begin{array}{ll} S_{a}\left(t,P_{m}\right) & =\arg\max\left|\int_{p_{m}}\left[\sigma_{s}\sin c\left[\left(t-\frac{R_{0}}{c}\right)\right]\exp\left(-j2\pi \frac{\left(v-v_{k}\right)}{c}p_{m}\frac{f_{c}}{f_{c}+f}\Delta f_{m}\right)\right.\right.\\ & \left.\left.\cdot \exp\left(-j2\pi \frac{\left(a-a_{0}\right)}{2c}p_{m}^{2}{\left(\frac{f_{c}}{f_{c}+f}\right)}^{2}\Delta f_{m}\right)\right]\exp\left(-j2\pi \frac{v}{c}p_{m}f_{c}\right)\exp\left(-j2\pi fp_{m}\right)dp_{m}\right| \end{array}$$

When *v* = *v_k_*, *a* = *a*_0_, Equation (23) reaches its maximum value. Equation (23) can be further written as
(24)$$S_{a}\left(t,P_{m}\right)=\int_{p_{m}}\sigma_{s}\sin c\left[\left(t-\frac{R_{0}}{c}\right)\right]\exp\left(-j2\pi \frac{v}{c}p_{m}f_{c}\right)\cdot \exp\left(-j2\pi fp_{m}\right)dp_{m}$$

In Equation (24), it is observed that the target’s energy is concentrated into a single-range Doppler cell for a moving target with constant acceleration when using FH signals. The searching scope for *a* is [*a*_min_, *a*_max_].

Solving Equation (24), the velocity estimation result is represented as
(25)$$v_{0}=\frac{c}{f_{c}}f$$

In Equation (25), it can be seen that the estimated target velocity *v*_0_ is irrelevant to the hopping frequency Δ*f*_m_, the target’s acceleration *a*, and the slow time *p_m_* compared to the result in Equation (16), and it is a constant value. The target’s energy concentrates into a one-resolution cell. Thus, DFM caused by the FH characteristic is compensated for, which means that long-time coherent integration for FH signals is achieved.

The summarized framework of the proposed method is given in Algorithm 1.
**Algorithm 1 Summary: The main step of the proposed method****1: Input:** Original reference signal *s*_ref_(*t*,*t*_m_) and surveillance signal *s*_sur_(*t*,*t*_m_), searching scope of target’s acceleration [*a*_min_, *a*_max_].**2: Fourier transform:** Apply FT to *s*_ref_(*t*,*t*_m_) and *s*_sur_(*t*,*t*_m_) with respect to fast time *t* to obtain *S*_ref_(*f*,*t*_m_) and *S*_sur_(*f*,*t*_m_).**3: Pulse compression:** Apply pulse compression in frequency domain to obtain *S*(*f*,*t*_m_) by Equation (5).**4: Range curve correction:** Apply GKT to *S*(*f*,*t*_m_) and achieve *S*(*f*,*τ_m_*) by Equation (7).**5: DMF-I correction:** Go through each *a* in [*a*_min_, *a*_max_].6: **For** *a* = *a*_min_, …, *a*_max_7: Multiply the acceleration correction factor *H*(*τ_m_*) and *S*(*f*,*τ_m_*) to obtain *S_a_*(*f*,*τ_m_*) by Equation (9).8: Preserve the peak of *S_a_*(*f*,*τ_m_*).9: **End****10: Find the matched acceleration:** Find the maximum peak value, the corresponding *a* is the estimated acceleration.**11: Range walk correction:** Apply GKT to *S_a_*(*f*,*τ_m_*) and achieve *S_a_*(*f*,*p_m_*) by Equation (13).**12: Inverse Fourier Transform:** Convert *S_a_*(*f*,*p_m_*) to *p_m_*-*t* domain form to obtain *s_a_*(*t*,*p_m_*) by Equation (14).**13: Compensation factor calculation:** Attribute the velocity compensation factor into the rotation factor of FT to obtain a new rotation factor *H_w_*(*p_m_*,*k*) and design a new acceleration correction factor *H_a_*(*p_m_*) according to the estimated acceleration *a*. **14: DMF-II correction:** Multiply *s_a_*(*t*,*p_m_*) and *H_a_*(*p_m_*), then use *H_w_*(*p_m_*,*k*) as a new rotation factor of FT to transform the multiplication result into the slow time frequency domain to obtain the coherent integration result *S_a_*(*t*,*P_m_*).

## 4. Simulation Results

In this section, numerical simulations are applied to verify the effectiveness of the proposed method. To realize the comparison experiments for a single target, serval methods, for example, the conventional range-Doppler (RD) method, the GKT method, and the DFM-II compensation method are presented as well. In this simulation, the FH pulse signals with a frequency hopping bandwidth of 100 MHz were employed as the illuminator of opportunity for the PBR system. By channelized receiving, both the reference signal and the surveillance signal were sampled with 5 MHz and observed for 0.12 s. Related simulation parameters are listed in Table 1.

### 4.1. SNR Analysis

Firstly, to show the SNR improvement in the proposed method, the integration results of each processing step are displayed in Figure 3.

In Figure 3a, it can be observed that the main peak of the target echo is distributed from the 150th to 154th range cell, which was caused by the RC and RW effects. According to the velocity dimension in Figure 3b, the energy of the target echo is distributed at around −600 m/s, which was caused by the DFM-I effect corresponding to the fourth phase item in Equation (6). Additionally, part of the target’s energy was spread to a larger Doppler frequency scope, which was caused by the DFM-II effect corresponding to the fifth and sixth phased items in Equation (6). Then, in Figure 3c, it can be observed that the main peak of the target echo is distributed from the 150th to 152nd range cell, indicating that it performed better compared to that in Figure 3a since the RC effect was removed. According to the velocity dimension in Figure 3d, the result is deteriorated compared to that in Figure 3b due to the variable substitution results in a more complex coupled problem related to the FH characteristics. In Figure 3e, it can be seen that there was little SNR improvement compared to Figure 3c. Since the DFM-I effect was corrected, the SNR output in Figure 3f was about 1 dB better compared to Figure 3d. In Figure 3g, it can be seen that the main peak of the target echo is concentrated in the 150th range cell since the RW effect was corrected. Due to the SNR improvement in the range dimension, the energy level of the target in the velocity domain was also increased. In Figure 3i, it can be seen that the main peak of the target echo was still concentrated in the 150th range cell, and the SNR output was 6 dB better than that in Figure 3g since the SNR improvement occurred in the velocity dimension. Due to the DFM-II effect correction, the main peak of the target echo is concentrated at around −600 m/s in Figure 3j. The ideal gain was log_10_(1200 × 10*^−^*^6^ × 5 × 10^6^) = 37.8 dB, while the actual gain in the simulation was around 32 dB. The gain difference was caused by the remaining DFM-II results based on the FH characteristics.

### 4.2. Computational Complexity

In this part, the computational complexity of the major step in the proposed method is analyzed. Suppose that N represents the sampling number in the fast time domain, and M represents the sampling number in the slow time domain. In the proposed method, fast FT (FFT) operation on the reference signal and surveillance signal requires 2MNlog_2_N multiplications. Regarding pulse compression operation in the frequency domain, it requires MN multiplications. In this paper, chirp-z transform (CZT) was applied to realize Keystone transform. To realize RC and RW correction, it was required to apply CZT twice. The total multiplications in this step are 2N(L + 2M + 1.5Llog_2_L + 0.5Mlog_2_M), where L ≥ 2M − 1 and L is an integer power of 2. Additionally, acceleration *a* needs to be estimated in both the DFM-I and DFM-II correction operations; the total multiplications in this step are 2Mlog_2_M. To compensate for the phase error caused by the target’s velocity in DFM-II, the velocity compensation factor is applied to the rotation factor of FT. Then, applying FFT to the RW correction result with respect to the fast time frequency *f* requires 2MNlog_2_N multiplications. Since the symmetry and periodicity characteristics of the rotation factor are broken, the FFT algorithm is not suitable for use anymore. Thus, the total multiplications in this step are 2MN^2^. To sum up, the computational complexities of this major step are displayed in Table 2.

### 4.3. Integration Results

Before illustrating the effectiveness of the proposed method, numerical analysis of RC, RW, and two kinds of DFM effects is presented. According to the simulation parameters, the range resolution cell and the Doppler resolution cell were 15 m and 8.33 Hz, respectively. The target moved 73.44 m across a 4.896 range resolution cell within the observation time, resulting in RC and RW. Additionally, the velocity varied from −600 m/s to −624 m/s, which spanned a 19.2077 Doppler resolution cell (corresponding to the center carrier frequency) within the observation time, resulting in DFM-I. The maximum and the minimum carrier frequency were 1.05 GHz and 0.95 GHz. Thus, the Doppler frequency of the target with an initial velocity of −600 m/s was −3800 Hz (corresponding to a carrier frequency of 0.95 GHz), whereas the Doppler frequency of the target with a final velocity of −624 m/s was −4368 Hz (corresponding to a carrier frequency of 1.05 GHz). The Doppler scope of the target was 568 Hz, which spanned a 68.1873 Doppler resolution cell within the observation time, resulting in DFM-II. To sum up, long-time integration for maneuvering the target using FH signals as radar sources suffered from a serious migration problem. Improving the detection performance required eliminating and compensating for RC, RW, and two kinds of DFM effects. Thus, the weak target echo was integrated coherently. The four mentioned methods were applied, and the results are shown in Figure 4.

Figure 4a–c show the integration results of the conventional RD method. It can be observed that the energy of the target echo was distributed into several range and Doppler resolution cells due to a serious migration effect. The estimated range cell and velocity values were 154 and −619.4 m/s, which do not correspond to the initialized value. The average energy level of the side lobe caused by the FH characteristics of the transmitting signal was about −32 dB, which is 2 dB lower than the main lobe of the target echo. Figure 4d–f show the integration results of the GKT method. Since the RC, RW, and DFM-I effects were eliminated and compensated for, the energy level was still distributed into several Doppler resolution cells due to the DFM-II effects. The estimated range cell information corresponds to the initialized value, whereas the estimated velocity value is inconsistent with the initialized value. The average energy level of the side lobe caused by the FH characteristics of the transmitting signal was about −29 dB, which is 7 dB lower than the main lobe of the target echo. Figure 4g–i show the integration results of the DFM-II compensation method. Only the DFM-II effects were compensated for, the energy level was still distributed into several range and Doppler resolution cells. Neither the range cell nor the velocity is inconsistent with the initialized value. The average energy level of the side lobe caused by the FH characteristics of the transmitting signal was about −36 dB, which is 10 dB lower than the main lobe of the target echo. Figure 4j–l show the integration results of the proposed method. It can be seen that the target energy was well-focused within most of the observation time since RC, RW and two kinds of DFM effects were eliminated and compensated for. The average energy level of the side lobe caused by the FH characteristics of the transmitting signal was about −34 dB, which is 18 dB lower than the main lobe of the target echo. The simulation results in Figure 4 illustrate that the proposed method can deal with the long-time coherent integration problem for FH signals with better performance compared to the other three methods.

It can be seen in Figure 4 that the simulated target could still be detected by using the parameter in Table 1. To further show the advantages of the proposed method, the SNR input of the target echo was decreased to −26 dB. The integration results of the four methods are displayed in Figure 5.

It can be seen in Figure 5 that the target could not be detected by the other three mentioned methods since the energy of the target echo was distributed into several range and velocity cells and the main peak of the target echo was submerged by the thermal noise. Since the RC, RW, DFM-I and DFM-II effects were compensated for and eliminated, the target echo was integrated coherently within most of the integration time. The target could still be detected by the proposed method. The simulation results illustrate that the proposed method can deal with the long-time integration problem for FH signals with low SNR input.

### 4.4. Performance Analysis

To further illustrate the performance of the proposed method, the input–output SNR performance and detection ability were assessed. The target’s motion parameters and the PBR system parameters were the same as the parameters listed in Table 1. For the input–output SNR performance experiment, the input SNRs tested varied from −30 to 10 dB with a step of 1 dB. The results of the four mentioned methods are shown in Figure 6a. As for the detection ability experiment, when the SNR output was 12 dB higher than the noise platform, the target was considered detected. The probability of detection of the four methods with different SNR inputs were calculated by 100 repetitions of Monte Carlo trials. The results are shown in Figure 6b.

In Figure 6, it can be observed that the proposed method achieved a better integration performance compared to the other three methods. Some issues were noted: (1) Compared to the other three methods, the proposed method achieved a better input–output and a higher detection ability. The reason is that the proposed method solves all the migration problems, whereas the other methods solve some of the migration problems. (2) The detection abilities of the GKT method and the DFM-II compensation method were 6 dB and 2.5 dB better than the conventional RD method, which illustrates that the RW, RC and DFM-I effects were more serious than the DFM-II effects in this simulation condition. (3) Compared to the ideal integration results, the proposed method still loses 6 dB in performance due to the phase compensation error caused by the estimated offset of the velocity and the acceleration. (4) Although the detection probability of the other three methods exceeded 0.9 when the SNR input was bigger than −8 dB, the target’s energy was distributed into serval resolution cells, which may result in parameter estimation error and false target detection.

## 5. Conclusions

Aiming at the coherent integration problem of FH signals, a novel method is proposed in this paper. In this method, RC and DFM are eliminated in the first step, and RW is eliminated in the second step. DFM caused by the FH characteristics is compensated for in the last step. Since the migration effects involving RC, RW and two kinds of DFM are eliminated and compensated for, the weak target echo can be integrated coherently within the observation time, which solves the coherent integration problem of FH signals. The simulation results and performance analysis illustrate that the proposed method can achieve better integration performance compared to the other three methods. Although the proposed method can achieve satisfactory performance compared to the existing methods, it suffers from a high computation cost in real applications. Studying the fast compensation implement algorithm and the efficient parameter estimation algorithm are interesting topics and will be part of our future study. Additionally, the motion model mismatch problem in real-world applications is also a key research point.

## Figures and Tables

**Figure 1 sensors-24-06236-f001:**
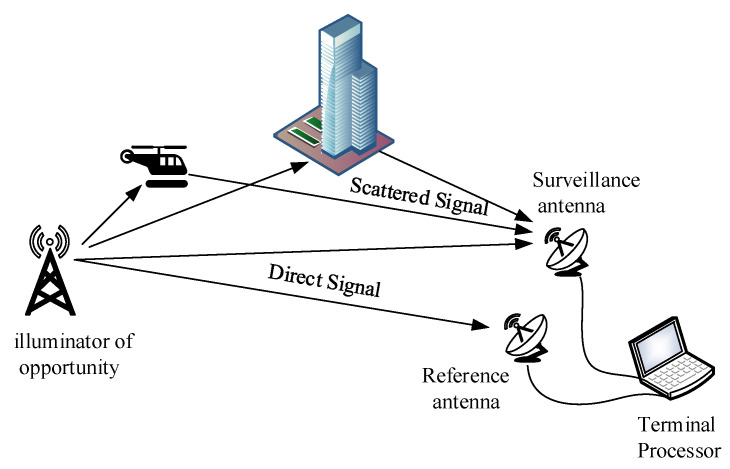
System geometry of passive bistatic radar.

**Figure 2 sensors-24-06236-f002:**
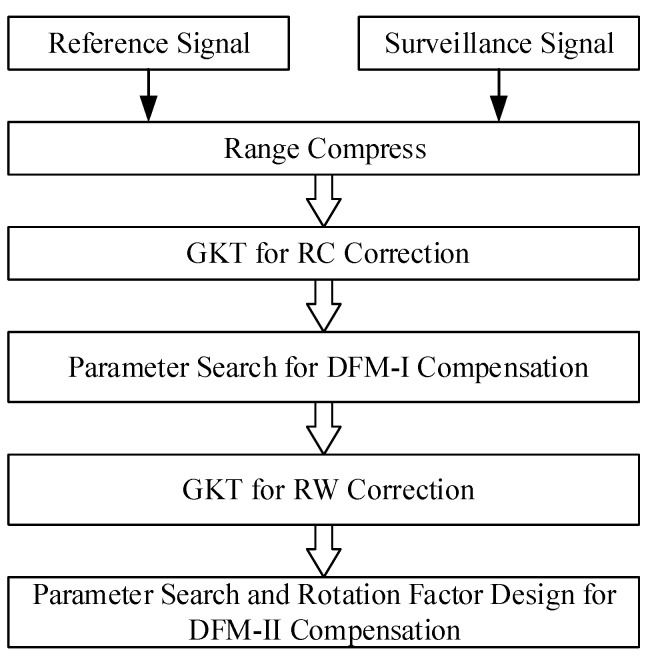
Flowchart of the proposed method.

**Figure 3 sensors-24-06236-f003:**
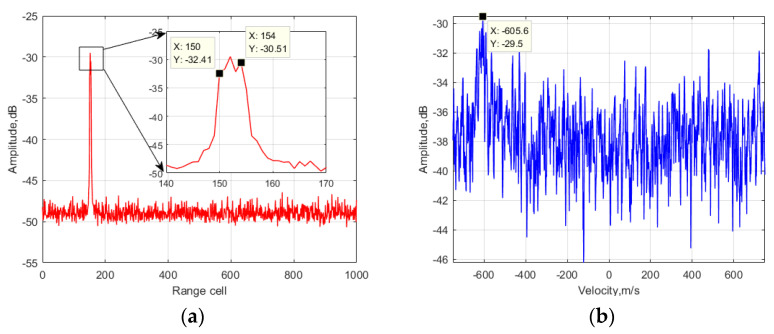
Output of the proposed method in different steps: (**a**) first step (original processing in the range dimension), (**b**) first step (original processing in the velocity dimension), (**c**) second step (after RC correction in the range dimension), (**d**) second step (after RC correction in the velocity dimension), (**e**) third step (after DFM-I correction in the range dimension), (**f**) third step (after DFM-I correction in the velocity dimension), (**g**) fourth step (after RW correction in the range dimension), (**h**) fourth step (after RW correction in the velocity dimension), (**i**) fifth step (after DFM-II correction in the range dimension), (**j**) fifth step (after DFM-II correction in the velocity dimension).

**Figure 4 sensors-24-06236-f004:**
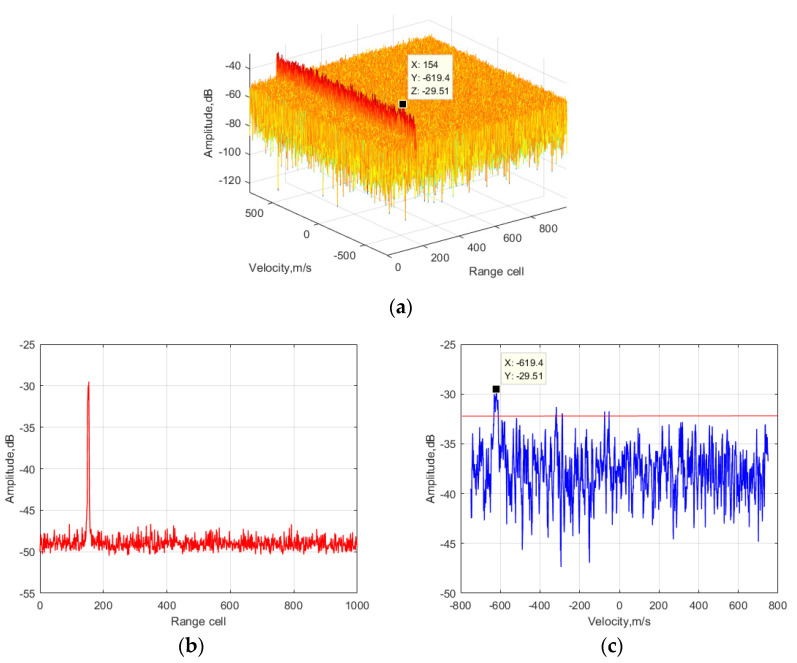
Output of the coherent integration processing method: (**a**) output of the conventional RD method; (**b**) output of the conventional RD method in the range dimension; (**c**) output of the conventional RD method in the velocity dimension; (**d**) output of the GKT method; (**e**) output of the GKT method in the range dimension; (**f**) output of the GKT method in the velocity dimension; (**g**) output of the DFM-II compensation method; (**h**) output of DFM-II compensation method in the range dimension; (**i**) output of the DFM-II compensation method in the velocity dimension; (**j**) output of the proposed method; (**k**) output of the proposed method in the range dimension; (**l**) output of the proposed method in the velocity dimension.

**Figure 5 sensors-24-06236-f005:**
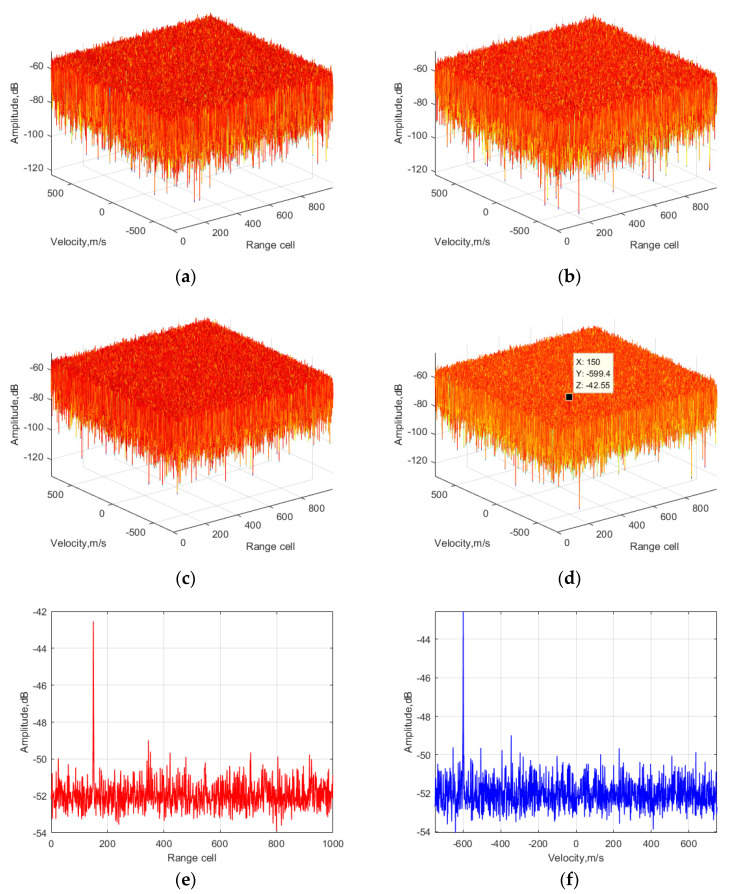
Output of the coherent integration processing method with low SNR input: (**a**) output of the conventional RD method; (**b**) output of the GKT method; (**c**) output of the DFM-II compensation method; (**d**) output of the proposed method; (**e**) output of the proposed method in the range dimension; (**f**) output of the proposed method in the velocity dimension.

**Figure 6 sensors-24-06236-f006:**
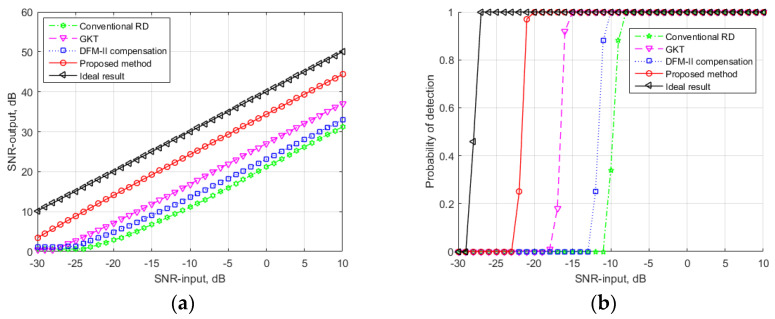
Performance of the coherent integration processing method: (**a**) input–output SNR performance; (**b**) detection ability.

**Table 1 sensors-24-06236-t001:** Simulation parameters.

Parameters	Values
Center frequency	1 GHz
Frequency hopping bandwidth	100 MHz
Signal bandwidth	5 MHz
Sampling frequency	5 MHz
Observation time	0.12 s
Pulse duration	0.5 µs
Pulse repetition time	1 µs
Pulse number	1200
Target’s initial range cell	150
Target’s initial velocity	−600 m/s
Target’s acceleration	−200 m/s^2^
Target’s SNR	−10 dB

**Table 2 sensors-24-06236-t002:** Computational complexities.

Step	Computational Complexities
FFT operation on reference and surveillance signals	2MNlog_2_N
Pulse compression operation in frequency domain	MN
CZT in RC and RW correction	2N(L + 2M + 1.5Llog_2_L + 0.5Mlog_2_M)
Acceleration estimation in DFM-I and DFM-II	2Mlog_2_M
FFT operation on correction result	2MNlog_2_N
Velocity compensation in DFM-II	2MN^2^

## Data Availability

No new data were created or analyzed in this study, so data sharing is not applicable.

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
