# Peer review of "Long-Time Coherent Integration Method for Passive Bistatic Radar Using Frequency Hopping Signals"

_sensors, 2024, doi:10.3390/s24196236_

Round 1
Reviewer 1 Report
Comments and Suggestions for Authors
This paper by Chen, G. et al. presents a method for long time coherent integration in radars using frequency hopping signals considering accelerated targets. Although, in my opinion, the addressed topic is of interest and the method presented might be relevant to be considered for publication, the English of the paper is in general difficult to follow. Therefore, I believe the English should be reviewed in order to make the paper clearer. For example, the use of “except” (page 1 line 9, and page 2 line 59) is not suitable (maybe change to “apart from”), or in page 5 line 145 “does Equation (15) reach its…” should be “Equation (15) reaches its…”.
There are also several typos, for example: in page 4 line 125, I think “RW” should be “RC”; there are two Equations (24); or in Table 1, “Center Frequency” is duplicated.
Additionally, the authors could consider the following comments to improve the quality, clarity and relevance of the manuscript:
- Why is the focus of the paper in passive bistatic radar applications when the presented method seems also to be valid for monostatic active radars using frequency hopping or stepped frequency? Besides, it could be clarified which illuminators of opportunity for passive radar use this kind of waveform in order to better justify the motivation of this paper.
- The contributions of this paper could be better described in comparison with your previous work [36].
- In the signal model of Eq. (1), I think that the exponential term is usually considered positive. Besides, I think that it should be commented and justified that a stop-and-go approximation is assumed for Eq. (2). Another assumption that should be made explicit is that the target reflectivity is assumed constant during the integration time (despite the change in frequency and movement of the target).
- In the simulation parameters, I think that considering a target acceleration of -200 m/s2 is not a common exemplary scenario for passive radar applications. Therefore, I would suggest to consider a more common target acceleration or to actually assess the effect of different target velocities and acceleration in the performance of the integration methods in order to justify in which cases it would be suitable to use the proposed method despite its additional computational complexity. Besides, it should be made explicit that a constant bistatic acceleration is considered in the simulations, which is a very specific situation. Even a linear target trajectory with constant velocity might induce a time-dependent bistatic acceleration in a bistatic radar scenario, so it would be relevant to analyze the sensitivity of the proposed method to acceleration changes during the integration time (for example considering a linear accelerated trajectory in Cartesian coordinates).
- Regarding figure 3, I think that a surface plot showing the computed Range-Doppler maps, and maybe adding the range and Doppler cuts or some zoomed views, would be clearer to see the sidelobes and spread of target energy in several resolution cells.
- In page 10, line 263, I think that the fourth point, i.e 4), should be rewritten in a clearer way.
- I believe that adding experimental results would be of great interest and relevance. If they are not added, in the section 4 title, “Experimental” should be removed.
As stated in my comments and suggestions for authors, I believe that the English of the paper should be reviewed in order to improve clarity.
Reviewer 2 Report
Comments and Suggestions for Authors
The paper presents a novel and potentially impactful method for handling frequency hopping signals in PBR systems. However, to meet the high standards expected for publication, it requires major/minor revisions to clarify the methodology, address computational and noise-related challenges, and provide a more thorough analysis of the results.
1:The abstract is comprehensive but lacks specificity in describing the contributions and novelty of the work. It should be more explicit about the main achievements and the context in which the proposed method excels compared to existing techniques.
2:The introduction briefly mentions various challenges like range walk (RW), range curve (RC), and Doppler frequency migration (DFM) but does not provide sufficient background on why these challenges are significant in the context of FH signals. Including a more thorough explanation of these issues would strengthen the reader's understanding of the problem being addressed.
3:The methodology section introduces a novel coherent integration method that addresses RC, RW, and two types of DFM. However, the explanation of the proposed algorithm (e.g., the application of the second-order Keystone Transform and DFM compensation) could be clearer. The step-by-step process should be broken down further with intermediate results or examples to improve readability and understanding.
4:The handling of the second-order DFM correction is complex, and the paper should address the computational cost and potential trade-offs involved. A discussion of the algorithm's efficiency and scalability in real-world applications is necessary.
5:The results section presents a comparison between the proposed method and other techniques, such as the Generalized Keystone Transform (GKT). While the results clearly show the superiority of the proposed method in terms of energy concentration, the paper should discuss potential limitations or scenarios where the method might underperform, such as in highly cluttered environments or with low SNR.
6:The impact of noise on the proposed method’s performance is not deeply analyzed. The performance under different noise levels should be explored, and how the method handles noise in the context of Doppler frequency migration should be explicitly discussed.
7:The analysis provides a good comparative overview, but the lack of analytical derivation for the performance metrics (e.g., SNR improvement) limits the robustness of the conclusions. Incorporating more mathematical analysis or theoretical justification for the observed improvements would strengthen the section.
8:The conclusion effectively summarizes the findings but could benefit from a more detailed discussion on future work, particularly on how the method could be extended or adapted to other signal types or radar systems.
9:The references are relevant but somewhat limited in scope. The inclusion of recent advances in radar signal processing, particularly those dealing with range ambiguity and multi-frequency integration, would enrich the context of the work. It is recommended to include the following references to broaden the scope and relevance of the study:
"An advanced scheme for range ambiguity suppression of spaceborne SAR based on blind source separation" - This reference would add value by discussing advanced techniques in range ambiguity suppression, which is relevant when considering the RC and RW corrections in the proposed method.
"Multi-Frequency Coherent Integration Target Detection Algorithm for Passive Bistatic Radar" - Including this reference will provide a comparative basis for multi-frequency integration techniques, further emphasizing the novelty and effectiveness of the proposed method in handling frequency hopping signals.
10:The conclusion does not sufficiently address potential real-world challenges or the next steps for transitioning the method from simulation to practical implementation.
11:The proposed method’s sensitivity to model mismatches, such as inaccuracies in the acceleration or velocity estimation, should be discussed. This is particularly important for practical deployment where the target motion model might deviate from the assumptions made in the paper.
Reviewer 3 Report
Comments and Suggestions for Authors
The abbreviations in the titles could be replaced by the full names and the abbreviations in brackets, because there is enough space. This improves the readability. The method of your Keystone interpolation(chirp-z,sinc?) and related losses are not mentioned. The figures in 3 could be replaced by several RD maps with acceleration as parameter. This shows the power distribution in a better way than for the one-dimensional range and Doppler cuts. A very nice article is "Long Coherent Integration in Passive Radar Systems Using Super-Resolution Sparse Bayesian Learning", which you could add as reference. The first three steps are the same, the forth step is DFM-II correction for the frequency stepping. You could point out the necessary computational effort compared to the normal Doppler FFT. Figure 2 could also be improved, because the sequence is different. Additionally you could add the Keystone interpolation factors and compensation equations in the flow chart.
Round 2
Reviewer 2 Report
Comments and Suggestions for Authors
accept